# PRUNED GRAPH SCATTERING TRANSFORMS

**Vassilis N. Ioannidis** *
Dpt. of Electrical and Computer Engineering
Univ. of Minnesota
Minneapolis, MN, USA
*ioann006@umn.edu*

**Siheng Chen**
Mitsubishi Electric Research Laboratories
Cambridge, MA, USA
`schen@merl.com`

**Georgios B. Giannakis**
Dpt. of Electrical and Computer Engineering
Univ. of Minnesota
Minneapolis, MN, USA
*georgios@umn.edu*

## ABSTRACT

Graph convolutional networks (GCNs) have achieved remarkable performance in a variety of network science learning tasks. However, theoretical analysis of such approaches is still at its infancy. Graph scattering transforms (GSTs) are non-trainable deep GCN models that are amenable to generalization and stability analyses. The present work addresses some limitations of GSTs by introducing a novel so-termed pruned (p)GST approach. The resultant pruning algorithm is guided by a graph-spectrum-inspired criterion, and retains informative scattering features on-the-fly while bypassing the exponential complexity associated with GSTs. It is further established that pGSTs are stable to perturbations of the input graph signals with bounded energy. Experiments showcase that i) pGST performs comparably to the baseline GST that uses all scattering features, while achieving significant computational savings; ii) pGST achieves comparable performance to state-of-the-art GCNs; and iii) Graph data from various domains lead to different scattering patterns, suggesting domain-adaptive pGST network architectures.

## 1 INTRODUCTION

The abundance of graph-structured data calls for advanced learning techniques, and complements nicely standard machine learning tools that cannot be directly applied to irregular data domains. Permeating the benefits of deep learning to the graph domain, graph convolutional networks (GCNs) provide a versatile and powerful framework to learn from complex graph data (Bronstein et al., 2017). GCNs and variants thereof have attained remarkable success in social network analysis, 3D point cloud processing, recommender systems and action recognition. However, researchers have recently reported inconsistent perspectives on the appropriate designs for GCN architectures. For example, experiments in social network analysis have argued that deeper GCNs marginally increase the learning performance (Wu et al., 2019), whereas a method for 3D point cloud segmentation achieves state-of-the-art performance with a 56-layer GCN network (Li et al., 2019). These 'controversial' empirical findings motivate theoretical analysis to understand the fundamental performance factors and the architecture design choices for GCNs.

Aiming to bestow GCNs with theoretical guarantees, one promising research direction is to study graph scattering transforms (GSTs). GSTs are non-trainable GCNs comprising a cascade of graph filter banks followed by nonlinear activation functions. The graph filter banks are mathematically designed and are adopted to scatter an input graph signal into multiple channels. GSTs extract scattering features that can be utilized towards graph learning tasks (Gao et al., 2019), with competitive performance especially when the number of training examples is small. Under certain conditions on the graph filter banks, GSTs are endowed with energy conservation properties (Zou & Lerman,

---

*This work was mainly done while V. N. Ioanndis was working at Mitsubishi Electric Research Laboratories.

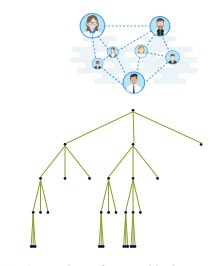 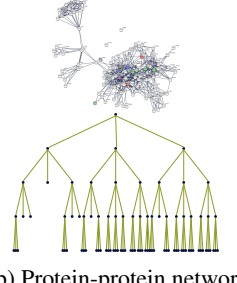 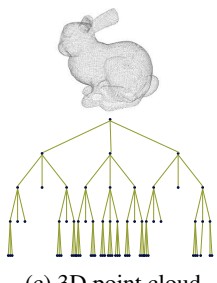

(a) Academic collaboration     (b) Protein-protein network     (c) 3D point cloud

Figure 1: Illustration of the same pGST applied to different graph datasets. Notice that for the social network (a) most of GST branches are pruned, suggesting that most information is captured by local interactions.

2019), as well as stability meaning robustness to graph topology deformations (Gama et al., 2019a). However, GSTs are associated with exponential complexity in space and time that increases with the number of layers. This discourages deployment of GSTs when a deep architecture is needed. Furthermore, stability should not come at odds with sensitivity. A filter's output should be sensitive to and "detect" perturbations of large magnitude. Lastly, graph data in different domains (social networks, 3D point clouds) have distinct properties, which encourages GSTs with domain-adaptive architectures.

The present paper develops a data-adaptive pruning framework for the GST to systematically retain important features. Specifically, the contribution of this work is threefold.

C1. We put forth a pruning approach to select informative GST features that we naturally term pruned graph scattering transform (pGST). The pruning decisions are guided by a criterion promoting alignment (matching) of the input graph spectrum with that of the graph filters. The optimal pruning decisions are provided on-the-fly, and alleviate the exponential complexity of GSTs.

C2. We prove that the pGST is stable to perturbations of the input graph signals. Under certain conditions on the energy of the perturbations, the resulting pruning patterns before and after the perturbations are identical and the overall pGST is stable.

C3. We showcase with extensive experiments that: i) the proposed pGSTs perform similarly and in certain cases better than the baseline GSTs that use all scattering features, while achieving significant computational savings; ii) The extracted features from pGSTs can be utilized towards graph classification and 3D point cloud recognition. Even without any training on the feature extraction step, the performance is comparable to state-of-the-art deep supervised learning approaches, particularly when training data are scarce; and iii) By analyzing the pruning patterns of the pGST, we deduce that graph signals in different domains call for different network architectures; see Fig. 1.

## 2 RELATED WORK

GCNs rely on a layered processing architecture comprising trainable graph convolution operations to linearly combine features per graph neighborhood, followed by pointwise nonlinear functions applied to the linearly transformed features (Bronstein et al., 2017). Complex GCNs and their variants have shown remarkable success in graph semi-supervised learning (Kipf & Welling, 2017; Veličković et al., 2018) and graph classification (Ying et al., 2018). To simplify GCNs, (Wu et al., 2019) has shown that by employing a single-layer linear GCN the performance in certain social network learning tasks degrades only slightly. On the other hand, (Li et al., 2019) has developed a 56-layer GCN that achieves state-of-the-art performance in 3D point cloud segmentation. Hence, designing GCN architectures guided by properties of the graph data is a highly motivated research question.

Towards theoretically explaining the success of GCNs, recent works study the stability properties of GSTs with respect to metric deformations of the domain (Gama et al., 2019b;a; Zou & Lerman, 2019). GSTs generalize scattering transforms (Bruna & Mallat, 2013; Mallat, 2012) to non-Euclidean

domains. GSTs are a cascade of graph filter banks and nonlinear operations that is organized in a tree-structured architecture. The number of extracted scattering features of a GST grows exponentially with the number of layers. Theoretical guarantees for GSTs are obtained after fixing the graph filter banks to implement a set of graph wavelets. The work in (Zou & Lerman, 2019) establishes energy conservation properties for GSTs given that certain energy-preserving graph wavelets are employed, and also prove that GSTs are stable to graph structure perturbations; see also (Gama et al., 2019b) that focuses on diffusion wavelets. On the other hand, (Gama et al., 2019a) proves stability to relative metric deformations for a wide class of graph wavelet families. These contemporary works shed light into the stability and generalization capabilities of GCNs. However, stable transforms are not necessarily informative, and albeit highly desirable, a principled approach to selecting informative GST features remains still an uncharted venue.

## 3 BACKGROUND

Consider a graph $\mathcal{G} := \{\mathcal{V}, \mathcal{E}\}$ with node set $\mathcal{V} := \{v_i\}_{i=1}^N$, and edge set $\mathcal{E} := \{e_i\}_{i=1}^E$. Its connectivity is described by the *graph shift matrix* $\mathbf{S} \in \mathbb{R}^{N \times N}$, whose $(n, n')$th entry $S_{nn'}$ is nonzero if $(n, n') \in \mathcal{E}$ or if $n = n'$. A typical choice for $\mathbf{S}$ is the adjacency or the Laplacian matrix. Further, each node can be also associated with a few attributes. Collect attributes across all nodes in the matrix $\mathbf{X} := [\mathbf{x}_1, \ldots, \mathbf{x}_F] \in \mathbb{R}^{N \times F}$, where each column $\mathbf{x}_f \in \mathbb{R}^N$ is a '*graph signal*.'

**Graph Fourier transform**. A Fourier transform corresponds to the expansion of a signal over bases that are invariant to filtering; here, this graph frequency basis is the eigenbasis of the graph shift matrix $\mathbf{S}$. Henceforth, $\mathbf{S}$ is assumed normal with $\mathbf{S} = \mathbf{V}\boldsymbol{\Lambda}\mathbf{V}^\top$, where $\mathbf{V} \in \mathbb{R}^{N \times N}$ forms the graph Fourier basis, and $\boldsymbol{\Lambda} \in \mathbb{R}^{N \times N}$ is the diagonal matrix of corresponding eigenvalues $\lambda_0, \ldots, \lambda_{N-1}$. These eigenvalues represent graph frequencies. The graph Fourier transform (GFT) of $\mathbf{x} \in \mathbb{R}^N$ is $\widehat{\mathbf{x}} = \mathbf{V}^\top \mathbf{x} \in \mathbb{R}^N$, while the inverse transform is $\mathbf{x} = \mathbf{V}\widehat{\mathbf{x}}$. The vector $\widehat{\mathbf{x}}$ represents the signal's expansion in the eigenvector basis and describes the graph spectrum of $\mathbf{x}$. The inverse GFT reconstructs the graph signal from its graph spectrum by combining graph frequency components weighted by the coefficients of the signal's graph Fourier transform. GFT is a tool that has been popular for analyzing graph signals in the graph spectral domain.

**Graph convolution neural networks**. GCNs permeate the benefits of CNNs from processing Euclidean data to modeling graph structured data. GCNs model graph data through a succession of layers, each of which consists of a graph convolution operation (graph filter), a pointwise nonlinear function $\sigma(\cdot)$, and oftentimes also a pooling operation. Given a graph signal $\mathbf{x} \in \mathbb{R}^N$, the graph convolution operation diffuses each node's information to its neighbors according to the graph shift matrix $\mathbf{S}$, as $\mathbf{S}\mathbf{x}$. The $n$th entry $[\mathbf{S}\mathbf{x}]_n = \sum_{n' \in \mathcal{N}_n} S_{nn'} x_{n'}$ is a weighted average of the one-hop neighboring features. Successive application of $\mathbf{S}$ will increase the reception field, spreading the information across the network. Hence, a $K$th order graph convolution operation (graph filtering) is

$$h(\mathbf{S})\mathbf{x} := \sum_{k=0}^K w_k \mathbf{S}^k \mathbf{x} = \mathbf{V}\widehat{h}(\boldsymbol{\Lambda})\widehat{\mathbf{x}} \tag{1}$$

where the graph filter $h(\cdot)$ is parameterized by the learnable weights $\{w_k\}_{k=0}^K$, and the graph filter in the graph spectral domain is $\widehat{h}(\boldsymbol{\Lambda}) = \sum_{k=0}^K w_k \boldsymbol{\Lambda}^k$. In the graph vertex domain, the learnable weights reflect the influences from various orders of neighbors; and in the graph spectral domain, those weights adaptively adjust the focus and emphasize certain graph frequency bands. GCNs employ various graph filter banks per layer, and learn the parameters that minimize a predefined learning objective, such as classification, or regression.

**Graph scattering transforms**. GSTs are the nontrainable counterparts of GCNs, where the parameters of the graph convolutions are selected based on mathematical designs. GSTs process the input at each layer by a sequential application of graph filter banks $\{h_j(\mathbf{S})\}_{j=1}^J$, an elementwise nonlinear function $\sigma(\cdot)$, and a pooling operator $U$. At the first layer, the input graph signal $\mathbf{x} \in \mathbb{R}^N$ constitutes the first scattering feature vector $\mathbf{z}_{(0)} := \mathbf{x}$. Next, $\mathbf{z}_{(0)}$ is processed by the graph filter banks and $\sigma(\cdot)$ to generate $\{\mathbf{z}_{(j)}\}_{j=1}^J$ with $\mathbf{z}_{(j)} := \sigma(h_j(\mathbf{S})\mathbf{z}_{(0)})$. At the second layer, the same operation is repeated per $j$. The resulting computation structure is a tree with $J$ branches at each non-leaf node; see also Fig. 2. The $\ell$th layer of the tree includes $J^\ell$ nodes. Each tree node at layer $\ell$ in the scattering transform is indexed by the path $p^{(\ell)}$ of the sequence of $\ell$ graph convolutions applied to the input

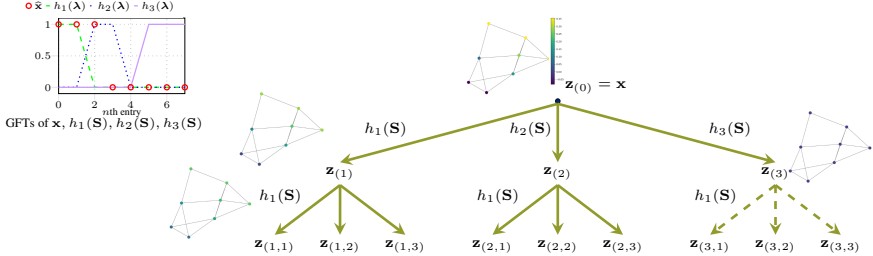

Figure 2: Scattering pattern associated with a pGST with $J = 3$ and $L = 3$. The dashed lines represent the pruned branches. The example of a graph signal and the GFTs of $\mathbf{x}$ and the filter banks are included as well. Note that the third filter $j = 3$ at $\ell = 1$ generates no output, i.e. $\mathbf{z}_{(3)} = \mathbf{0}$, and hence is pruned.

graph signal $\mathbf{x}$, i.e. $p^{(\ell)} := (j^{(1)}, j^{(2)}, \ldots, j^{(\ell)})$.[1] The scattering feature vector at the tree node indexed by $(p^{(\ell)}, j)$ at layer $\ell + 1$ is

$$\mathbf{z}_{(p^{(\ell)}, j)} = \sigma(h_j(\mathbf{S})\mathbf{z}_{(p^{(\ell)})}) \tag{2}$$

where the variable $p^{(\ell)}$ holds the list of indices of the parent nodes ordered by ancestry, and all path $p^{(\ell)}$ in the tree with length $\ell$ are included in the path set $\mathcal{P}^{(\ell)}$ with $|\mathcal{P}^{(\ell)}| = 2^\ell$. The nonlinear transformation function $\sigma(\cdot)$ disperses the graph frequency representation through the spectrum, and endows the GST with increased discriminating power (Gama et al., 2019a). By exploiting the sparsity of the graph, the computational complexity of (2) is $\mathcal{O}(KE)$, where $E = |\mathcal{E}|$ is the number of edges in $\mathcal{G}$.[2] Each scattering feature vector $\mathbf{z}_{(p^{(\ell)})}$ is summarized by an aggregation operator $U(\cdot)$ to obtain a scalar scattering coefficient as $\phi_{(p^{(\ell)})} := U(\mathbf{z}_{(p^{(\ell)})})$, where $U(\cdot)$ is typically an average or sum operator that effects dimensionality reduction of the extracted features. The scattering coefficient at each tree node reflects the activation level at a certain graph frequency band.

These scattering coefficients are collected across all tree nodes to form a scattering feature map

$$\boldsymbol{\Phi}(\mathbf{x}) := \left\{\{\phi_{(p^{(\ell)})}\}_{p^{(\ell)} \in \mathcal{P}^{(\ell)}}\right\}_{\ell=0}^{L} \tag{3}$$

where $|\boldsymbol{\Phi}(\mathbf{x})| = \sum_{\ell=0}^{L} J^\ell$. The GST operation resembles a forward pass of a trained GCN. This is why several works study GST stability under perturbations of $\mathbf{S}$ in order to understand the working mechanism of GCNs (Zou & Lerman, 2019; Gama et al., 2019a;b).

## 4 PRUNED GRAPH SCATTERING TRANSFORMS

While the representation power of GST increases with the number of layers, the computational and space complexity of the transform also increase exponentially with the number of layers due to its scattering nature. Hence, even if informative features are available at deeper GST layers, the associated exponential complexity of extracting such features is prohibitive with the existing GST architectures. On the other hand, various input data (social networks, 3D point clouds) may have distinct properties, leading to different GST feature maps. In some cases, only a few tree nodes in deep layers are informative; and in other cases, tree nodes in shallow layers carry significant information; see Fig. 1. This requires a customized GST to adaptively choose significant tree nodes.

Alleviating GST limitations, we introduce a *pruned graph scattering transform* (pGST) to systematically retain informative tree nodes without additional complexity. Our novel pGST alleviates the exponential complexity and adapts GST to different input data. Furthermore, pGST offer a practical mechanism to understand the architecture of GCNs. Based on the pruning patterns, the proposed pGST suggests when a deeper GCN is desirable, and when a shallow one will suffice. Pruning the wavelet packets has been traditionally employed for compression in image processing

---

[1]A tree node is fully specified by its corresponding path.

[2]Any analytical function $h(\mathbf{S})$ can be written as a polynomial of $\mathbf{S}$ with maximum degree $N - 1$ (Horn & Johnson, 2012).

applications (Xiong et al., 2002), where the pruning is guided by a rate-distortion optimallity criterion. In this work, we consider a graph spectrum inspired criterion. Intuitively, each tree node in the GSTs reflects a unique subband in the graph spectrum. When the subband of a tree node does not have a sufficient overlap with the the graph spectrum of a graph signal, this tree node cannot capture the property of this graph signal, and should be pruned.

For example, consider a smooth graph signal $\mathbf{x}$, where connected nodes have similar signal values, that has a sparse (low-rank) representation in the graph spectral domain that is, $\widehat{\mathbf{x}} := \mathbf{V}^\top \mathbf{x} \in \mathbb{R}^N$ and $[\widehat{\mathbf{x}}]_n = 0$ for $n \geq b$. The graph spectrum of the $j$th graph filtered output is then

$$\mathbf{V}^\top h_j(\mathbf{S})\mathbf{x} = \mathrm{diag}\left(\widehat{h}_j(\boldsymbol{\lambda})\right)\widehat{\mathbf{x}} = [\widehat{h}_j(\lambda_1)\widehat{x}_1, \widehat{h}_j(\lambda_2)\widehat{x}_2, \ldots, \widehat{h}_j(\lambda_N)\widehat{x}_N]^\top$$

where $\lambda_n$ is the $n$th eigenvalue of $\mathbf{S}$ and each frequency $\widehat{x}_n$ is weighted by the corresponding transformed eigenvalue $\widehat{h}_j(\lambda_n)$. Hence, if the support of the graph spectrum $\{\widehat{h}_j(\lambda_n)\}_n$ is not included in the support of $[\widehat{\mathbf{x}}]_n$ then the $j$th graph filter output will not capture any information; that is, $h_j(\mathbf{S})\mathbf{x} = \mathbf{0}_N$; see Fig. 2. Thus, identifying such graph filters and pruning the corresponding tree nodes will result to a parsimonius and thus computationally efficient GST.

**Pruning criterion**. Motivated by the aforementioned observation, we introduce a pruning criterion to select the scattering branches per tree node by maximizing the alignment between the graph spectrum of the graph filters and the scattering feature. At the tree node $p$, the optimization problem is

$$\max_{\{f_{(p,j)}\}_{j=1}^J} \quad \sum_{j=1}^J \left( \sum_{n=1}^N \left( \widehat{h}_j(\lambda_n)^2 - \tau \right) [\widehat{\mathbf{z}}_{(p)}]_n^2 \right) f_{(p,j)} \tag{4}$$
$$\text{s. t.} \quad f_{(p,j)} \in \{0,1\}, \quad j = 1, \ldots, J$$

where $\widehat{\mathbf{z}}_{(p)} := \mathbf{V}\mathbf{z}_{(p)}$ is the graph spectrum of the scattering feature vector $\mathbf{z}_{(p)}$; $\tau$ is a user-specific threshold; and, $f_{(p,j)}$ stands for the pruning assignment variable indicating whether node $(p, j)$ is active ($f_{(p,j)} = 1$) or it should be pruned ($f_{(p,j)} = 0$). The objective in (4) promotes retaining tree nodes that maximize the alignment of the graph spectrum of $\widehat{\mathbf{z}}_{(p)}$ with that of $\widehat{h}_j(\boldsymbol{\lambda})$. The threshold $\tau$ introduces a minimum spectral value to locate those tree nodes whose corresponding graph spectral response is small, i.e. $\widehat{h}_j(\lambda_n)^2 \ll \tau$. Note that criterion (4) is evaluated per tree node $p$, thus allowing for a flexible and scalable design.

The optimization problem in (4) is nonconvex since $f_{(p,j)}$ is a discrete variable. Furthermore, recovering $\widehat{\mathbf{z}}_{(p)}$ requires an eigendecomposition of the Laplacian matrix that incurs $\mathcal{O}(N^3)$ complexity. Nevertheless, by exploiting the structure in (4), we develop an efficient pruning algorithm that achieves the maximum of (4), as summarized in the following theorem.

**Theorem 1.** *The optimal pruning assignment variables* $\left\{ f_{(p,j)}^* \right\}_j$ *of* (4) *is given as follows*

$$f_{(p,j)}^* = \begin{cases} 1 & \text{if } \frac{\|\mathbf{z}_{(p,j)}\|^2}{\|\mathbf{z}_{(p)}\|^2} > \tau, \\ 0 & \text{if } \frac{\|\mathbf{z}_{(p,j)}\|^2}{\|\mathbf{z}_{(p)}\|^2} < \tau. \end{cases} \quad , \; j = 1, \ldots, J \tag{5}$$

The optimal variables $f_{(p,j)}^*$ are given by comparing the energy of the input $\mathbf{z}_{(p)}$ to that of the output $\mathbf{z}_{(p,j)}$ per graph filter $j$ that can be evaluated at a low complexity of $\mathcal{O}(N)$. Our pruning criterion leads to a principled and scalable algorithm to selecting the GST tree nodes to be pruned. The pruning objective is evaluated at each tree node $p$, and pruning decisions are made on-the-fly. Hence, when $f_{(p)}^* = 1$, tree node $p$ is active and the graph filter bank will be applied to $\mathbf{z}_{(p)}$, expanding the tree to the next layer; otherwise, the GST will not be expanded further at tree node $p$, which can result to exponential savings in computations. An example of such a pruned tree is depicted in Fig. 2. Evidently, the hyperparameter $\tau$ controls the input-to-output energy ratio. A large $\tau$ corresponds to an aggressively pruned scattering tree, while a small $\tau$ amounts to a minimally pruned scattering tree. The pGST is then defined as

$$\boldsymbol{\Psi}(\mathbf{x}) := \big\{ \phi_{(p)} \big\}_{p \in \mathcal{T}}$$

where $\mathcal{T}$ is the set of active tree nodes $\mathcal{T} := \{p \in \mathcal{P} | f_{(p)}^* = 1\}$.

Our pruning approach provides a concise version of GSTs and effects savings in computations as well as memory. Although the worst-case complexity of pGST is still exponential, a desirable complexity can be effected by properly selecting $\tau$. As a byproduct, the scattering patterns of pGSTs reveal the appropriate depths and widths of the GSTs for different input data; see also Fig. 1. The pruning approach so far is an unsupervised one, since no input data labels are assumed available.

## 5 STABILITY AND SENSITIVITY OF PGST

In this section, we prove the stability of pGST to perturbations of the input graph signal. To establish the ensuing results, we consider graph wavelets that form a frame with frame bounds $A$ and $B$ (Hammond et al., 2011). Specifically, for any graph signal $\mathbf{x} \in \mathbb{R}^N$, it holds that, $A^2\|\mathbf{x}\|^2 \leq \sum_{j=1}^{J} \|h_j(\mathbf{S})\mathbf{x}\|^2 \leq B^2\|\mathbf{x}\|^2$. In the graph vertex domain, the scalar frame bounds $A$ and $B$ characterize the numerical stability of recovering a graph signal $\mathbf{x}$ from $\{h_j(\mathbf{S})\mathbf{x}\}_j$. In the graph spectral domain, they reflect the ability of the graph filter bank to amplify $\mathbf{x}$ along each graph frequency. Tight frame bounds, satisfying $A^2 = B^2$, are of particular interest because such wavelets lead to enhanced numerical stability and faster computations (Shuman et al., 2015). The frame property of the graph wavelet plays an instrumental role in proving GST stability to perturbations of the underlying graph structure (Gama et al., 2019a;b; Zou & Lerman, 2019).

Consider a perturbed graph signal $\tilde{\mathbf{x}}$ given by

$$\tilde{\mathbf{x}} := \mathbf{x} + \boldsymbol{\delta} \in \mathbb{R}^N \tag{6}$$

where $\boldsymbol{\delta} \in \mathbb{R}^N$ is the perturbation vector. Such an additive model (6) may represent noise in the observed feature or adversarial perturbations. We are interested in studying how and under which conditions our pGST is affected by such perturbations. A stable transformation should have a similar output under small input perturbations.

Before establishing that our pGST is stable, we first show that GST is stable to small perturbations of the input graph signal[3].

**Lemma 1.** *Consider the GST $\boldsymbol{\Phi}(\cdot)$ with $L$ layers and $J$ graph filters; and suppose that the graph filter bank forms a frame with bound $B$, while $\mathbf{x}$ and $\tilde{\mathbf{x}}$ are related via* (6)*. It then holds that*

$$\frac{\|\boldsymbol{\Phi}(\mathbf{x}) - \boldsymbol{\Phi}(\tilde{\mathbf{x}})\|_2}{\sqrt{|\boldsymbol{\Phi}(\mathbf{x})|}} \leq \sqrt{\frac{\sum_{\ell=0}^{L}(B^2 J)^\ell}{\sum_{\ell=0}^{L} J^\ell}}\|\boldsymbol{\delta}\|_2 \ . \tag{7}$$

The squared difference of the GSTs is normalized by the number of scattering features in $\boldsymbol{\Phi}(\cdot)$, that is $|\boldsymbol{\Phi}(\mathbf{x})| = \sum_{\ell=1}^{L} J^\ell$. The bound in (7) relates to the frame bound of the wavelet filter bank. Notice that for tight frames with $B = 1$, then the normalized stability bound (7) is tight. Let $\tilde{\mathcal{T}}$ be the structure of the pruned tree for $\boldsymbol{\Psi}(\tilde{\mathbf{x}})$. The following lemma asserts that the pGST offers the same pruned tree for the original and the perturbed inputs.

**Lemma 2.** *Let $\tilde{\mathbf{z}}_p$ denote the perturbed scattering feature at the tree node $p$ and $\boldsymbol{\delta}_p := \mathbf{z}_p - \tilde{\mathbf{z}}_p$. If for all $p \in \mathcal{P}$ and $j = 1, \ldots, J$, it holds that*

$$\left|\|h_j(\mathbf{S})\mathbf{z}_p\|^2 - \tau\|\mathbf{z}_p\|^2\right| > \|h_j(\mathbf{S})\boldsymbol{\delta}_p\|^2 + \tau\left|\|\mathbf{z}_p\|^2 - \|\tilde{\mathbf{z}}_p\|^2\right|. \tag{8}$$

*Then, we obtain*

1. *The pruning transform will output the same tree for $\boldsymbol{\Psi}(\mathbf{x})$ and $\boldsymbol{\Psi}(\tilde{\mathbf{x}})$; that is, $\mathcal{T} = \tilde{\mathcal{T}}$; and,*
2. *With $g(\mathbf{x}) := \|h_j(\mathbf{S})\mathbf{x}\|^2 - \tau\|\mathbf{x}\|^2$, a necessary condition for (8) is*

$$|g(\mathbf{z}_p)| > g(\boldsymbol{\delta}_p) \ . \tag{9}$$

According to (9), Lemma 2 can be interpreted as a signal-to-noise ratio (SNR) condition because under $g(\boldsymbol{\delta}_p) > 0$, it is possible to write (9) as $|g(\mathbf{z}_p)|/g(\boldsymbol{\delta}_p) > 1$. Lemma 2 provides a per-layer and branch condition for pGST to output the same scattering tree for the original or the perturbed signal. By combining Lemmas 1 and 2, we arrive at the following stability result for the pGST network.

---

[3]Prior art deals with GST stability to structure perturbations (Gama et al., 2019a;b; Zou & Lerman, 2019).

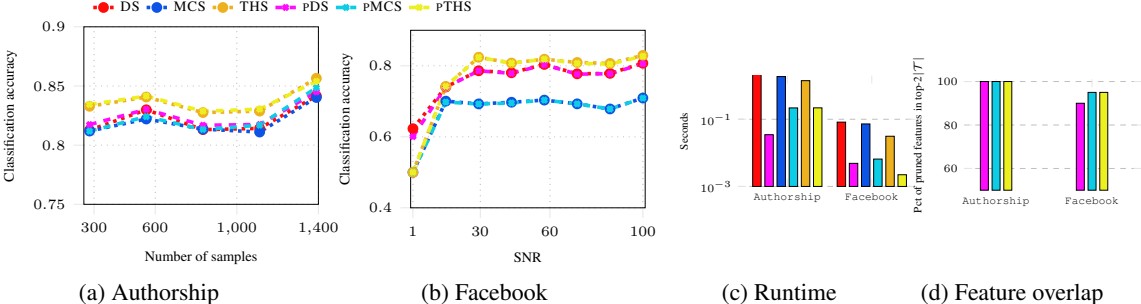

(a) Authorship     (b) Facebook     (c) Runtime     (d) Feature overlap

Figure 3: Classification accuracy against number of samples in the authorship attribution (a) and SNR in dB for source localization (b). Runtime comparison in seconds of the scattering transforms (c).

**Theorem 2.** *Consider the pGST transform $\mathbf{\Psi}(\cdot)$ with $L$ layers and $J$ graph filters; and suppose that the graph filter bank forms a frame with bound $B$, while $\mathbf{x}$ and $\tilde{\mathbf{x}}$ are related via* (6). *The pGST is stable to bounded perturbations $\boldsymbol{\delta}$, in the sense that*

$$\frac{\|\mathbf{\Psi}(\mathbf{x}) - \mathbf{\Psi}(\tilde{\mathbf{x}})\|_2}{\sqrt{|\mathbf{\Psi}(\mathbf{x})|}} \leq \sqrt{\frac{\sum_{\ell=0}^{L} F_\ell B^{2\ell}}{\sum_{\ell=0}^{L} F_\ell}} \|\boldsymbol{\delta}\|_2$$

*where $F_\ell := |\mathcal{P}^{(\ell)} \cup \mathcal{T}|$ is the number of active scattering features at layer $\ell$, and $|\mathbf{\Psi}(\mathbf{x})| = \sum_{\ell=0}^{L} F_\ell$ the number of retained scattering features.*

## 6 EXPERIMENTS

This section evaluates the performance of our pGST in various graph classification tasks. Graph classification amounts to predicting a label $y_i$ given $\mathbf{x}_i$ and $\mathbf{S}_i$ for the $i$th graph. Our pGST extracts $\mathbf{\Psi}(\mathbf{x}_i)$, which is utilized as a feature vector for predicting $y_i$. During training, the structure of the pGST $\mathcal{T}$ is determined, which is kept fixed during validation and testing. The parameter $\tau$ is selected via cross-validation. Our goal is to provide tangible answers to the following research questions.

    **RQ1** How does the proposed pGST compare to GST?
    **RQ2** How does pGST compare to state-of-the-art GCN approaches for graph classification?
    **RQ3** What are the appropriate scattering patterns for various graph data?

Appendix A includes additional experiments on ablation studies over the effect of the parameters $J, L, \tau$.

**pGST vs. GST**. To address RQ1, we reproduce the experiments of two tasks in (Gama et al., 2019a): authorship attribution and source localization. For the scattering transforms, we consider three implementations of graph filter banks: the diffusion wavelets (DS) in (Gama et al., 2019b), the monic cubic wavelets (MCS) in (Hammond et al., 2011) and the tight Hann wavelets (THS) in (Shuman et al., 2015).[4] The scattering transforms use $J = 5$ filters, $L = 5$ layers, and $\tau = 0.01$. The extracted features from GSTs are subsequently utilized by a linear support vector machine (SVM) classifier.

Authorship attribution amounts to determining if a certain text was written by a specific author. Each text is represented by a graph with $N = 244$, where words (nodes) are connected based on their relative positions in the text, and $\mathbf{x}$ is a bag-of-words representation of the text; see also (Gama et al., 2019b). Fig. 3 (a) reports the classification accuracy for the authorship attribution task as the number of training samples (texts) increases. GSTs utilize $\sum_{\ell=1}^{5} 5^\ell = 781$ scattering coefficients, while pGSTs rely only on $|\mathcal{T}| = 61$ for PDS, $|\mathcal{T}| = 30$ for PMCS, and $|\mathcal{T}| = 80$ for PTHS. Evidently, the proposed pGST achieves comparable performance as the baseline GST, whereas pGST uses only a subset of features ($12.8\%, 3.8\%$ and $10.2\%$ respectively). The SVM classifier provides a coefficient that weighs each scattering scalar. The magnitude of each coefficient shows the importance of the corresponding scattering feature in the classification. Fig. 3 (d) depicts the percentage of features

---

[4]PDS, PMCS, PTHS denote the pruned versions of these transforms.

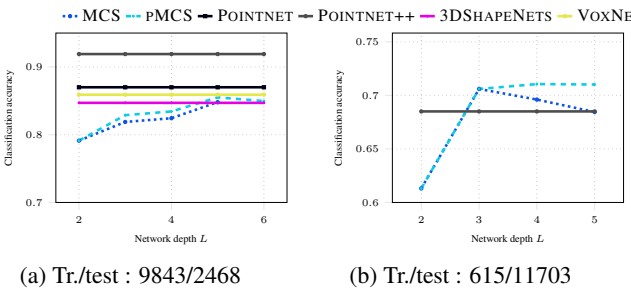

| Method | Data Set | | | |
|---|---|---|---|---|
| | ENZYMES | D&D | COLLAB | PROTEINS |
| **Kernel** | | | | |
| SHORTEST-PATH | 42.32 | 78.86 | 59.10 | 76.43 |
| WL-OA | 60.13 | 79.04 | 80.74 | 75.26 |
| **GNNs** | | | | |
| PATCHYSAN | – | 76.27 | 72.60 | 75.00 |
| GRAPHSAGE | 54.25 | 75.42 | 68.25 | 70.48 |
| ECC | 53.50 | 74.10 | 67.79 | 72.65 |
| SET2SET | 60.15 | 78.12 | 71.75 | 74.29 |
| SORTPOOL | 57.12 | 79.37 | 73.76 | 75.54 |
| DIFFPOOL-DET | 58.33 | 75.47 | **82.13** | 75.62 |
| DIFFPOOL-NOLP | 62.67 | 79.98 | 75.63 | 77.42 |
| DIFFPOOL | **64.23** | 81.15 | 75.50 | 78.10 |
| **Scattering** | | | | |
| GSC | 53.88 | 76.57 | 76.88 | 74.03 |
| GST | 59.84 | 79.28 | 77.32 | 76.23 |
| pGST (Ours) | 60.25 | **81.27** | 78.40 | **78.57** |

(a) Tr./test : 9843/2468     (b) Tr./test : 615/11703

Figure 4: 3D point cloud classification.     Table 1: Graph classification accuracy.

after prunning retained in the top-$2|\mathcal{T}|$ most important GST features given by the SVM classifier. It is observed, that although pGST does not take into account the labels, the retained features are indeed informative for classification.

Source localization amounts to recovering the source of a rumor given a diffused signal over a Facebook subnetwork with $N = 234$; see the detailed settings in (Gama et al., 2019b). Fig. 3 (b) shows the classification accuracy of the scattering transforms for increasing SNR in dB. In accordance to Lemma 1 and Theorem 2, both pGST and GST are stable for a wide range of SNR. Furthermore, the performance of pGST matches the corresponding GST, while the pGST uses only a subset of features. Fig. 3 (c) depicts the runtime of the different scattering approaches, where the computational advantage of the pruned methods is evident.

**Graph classification**. Towards answering RQ2, the proposed pGST is compared with the following state-of-the-art approaches.[5] The kernel methods shortest-path (Borgwardt & Kriegel, 2005), and Weisfeiler-Lehman optimal assignment (WL-OA) (Kriege et al., 2016); the deep learning approaches PatchySan (Niepert et al., 2016), GraphSage (Hamilton et al., 2017), edge-conditioned filters in CCNs (ECC) (Simonovsky & Komodakis, 2017), Set2Set (Vinyals et al., 2015), SortPool (Zhang et al., 2018), and DiffPool (Ying et al., 2018); and the geometric scattering classifier (GSC) (Gao et al., 2019). Results are presented with protein data sets D&D, Enzymes and Proteins, and the scientific collaboration data set Collab. Detailed description of the datasets is included in the Appendix. We perform 10-fold cross validation and report the classification accuracy averaged over the 10 folds. The gradient boosting classifier is employed for pGST and GST with parameters chosen based on the performance on the validation set. The graph scattering transforms use the MC wavelet with $L = 5$, $J = 5$ and $\tau = 0.01$. Table 1 lists the classification accuracy of the proposed and competing approaches. Even without any training on the feature extraction step, the performance of pGST is comparable to the state-of-the-art deep supervised learning approaches across all datasets. GST and pGST outperform also GSC, since the latter uses a linear SVM to classify the scattering features.

**Point cloud classification**. We further test pGST in classifying 3D point clouds. Given a point cloud, a graph can be created by connecting points (nodes) to their nearest neighbors based on their Euclidian distance. Each node is also associated with 6 scalars denoting its x-y-z coordinates and RGB colors. For this experiment, GSTs are compared against PointNet++ (Qi et al., 2017a;b), 3dShapeNets (Wu et al., 2015) and VoxNet (Maturana & Scherer, 2015), that are state-of-the-art deep learning approaches. Fig. 4 reports the classification accuracy for the ModelNet40 dataset (Wu et al., 2015) for increasing $L$. In Fig. 4 (a) 9,843 clouds are used for training and 2,468 for testing using the gradient boosting classifier; whereas, in Fig. 4 (b) only 615 clouds are used for training and the rest for testing using a fully connected neural network classifier with 3 layers. The scattering transforms use an MC wavelet with $J = 5$ for Fig. 4 (a) and $J = 9$ for Fig. 4 (b). Fig. 4 showcases that scattering transforms are competitive to state-of-the-art approaches, while pGST outperforms GST. This may be attributed to overfitting effects, since a large number of GST features is not informative. Furthermore, the exponential complexity associated with GSTs prevents their application for $L = 6$. Fig. 4 (b) shows that when the training data are scarce, GST and pGST outperform the PointNet++, which requires a large number of training data to optimize over the network parameters.

---

[5] For the competing approaches we report the 10-fold cross-validation numbers reported by the original authors; see also (Ying et al., 2018).

**Scattering patterns**. Towards answering RQ3, we depict the scattering structures of pGSTs, with a MC wavelet, $J = 3$ and $L = 5$, for the Collab, Proteins, and ModelNet40 datasets in Fig. 1. Evidently, graph data from various domains require an adaptive scattering architecture. Specifically, most tree nodes for the academic collaboration dataset are pruned, and hence most informative features are in the shallow layers. This is consistent with the study in (Wu et al., 2019), which experimentally shows that deeper GCNs do not contribute as much for social network data. These findings are further supported by the small-world phenomenon in social networks, which suggests the diameter of social networks is small (Watts & Strogatz, 1998). On the other hand, the tree nodes for a 3D point cloud are minimally pruned, which is in line with the work in (Li et al., 2019) that showcases the advantage of deep GCNs in 3D point clouds classification. For the protein datasets, additional experiments are performed in Appendix A.4 that corroborate the pGST insights regarding the required number of GCN layers.

## 7 CONCLUSIONS

This paper developed a novel approach to pruning the graph scattering transform. The proposed pGST relies on a graph-spectrum-based data-adaptive criterion to prune non-informative features on-the-fly, and effectively reduce the computational complexity of GSTs. Furthermore, when the input signal is perturbed, the stability of pGST is established. Experiments demonstrate that i) the performance gains of pGSTs relative to GSTs; ii) pGST is competitive in a variety of graph classification tasks; and (iii) graph data from different domains exhibit unique pruned scattering patterns, which calls for adaptive network architectures.

**Acknowlegdments.** This work was supported by Mitsubishi Electric Research Laboratories, the Doctoral Dissertation Fellowship of the University of Minnesota, and the NSF grants 171141, and 1500713.

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

| Dataset | Graphs | Features $F$ | Max number of nodes per graph |
|---------|--------|--------------|-------------------------------|
| Collab | 5000 | 1 | 492 |
| D&D | 1178 | 89 | 5748 |
| Enzymes | 600 | 3 | 126 |
| Proteins | 1113 | 3 | 620 |

Table 2: Dataset characteristics

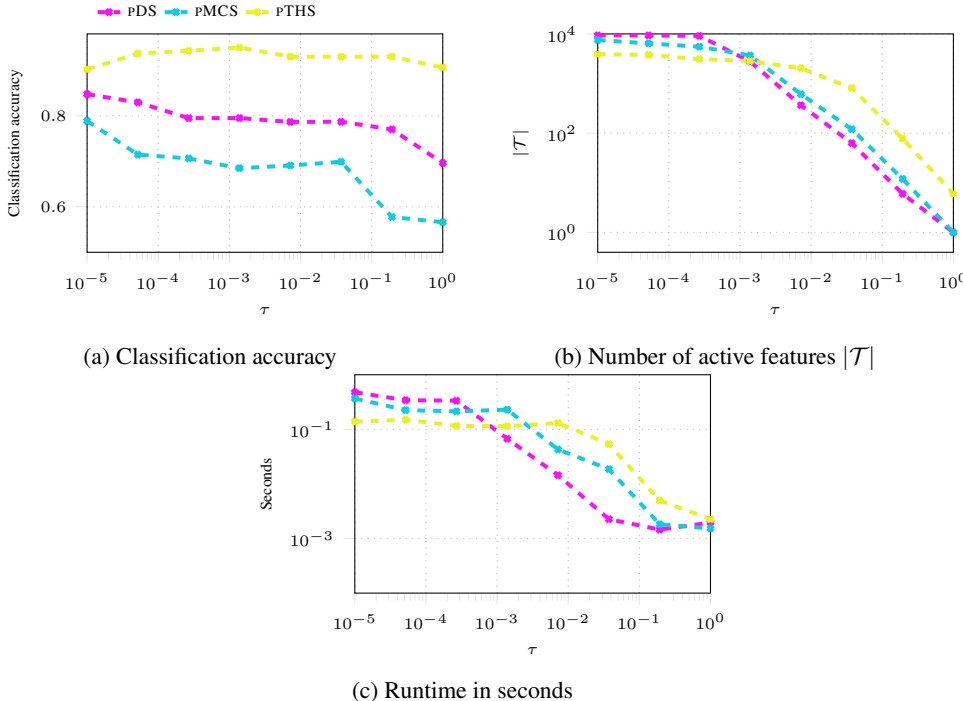

(a) Classification accuracy    (b) Number of active features $|\mathcal{T}|$

(c) Runtime in seconds

Figure 5: Performance of pGSTs for varying $\tau$.

# A    ADDITIONAL EXPERIMENTS

**Dataset characteristics**. The characteristics of the Datasets used in Table 1 are shown in Table 2. Notice that the nodes in the Collab dataset did not have any features, and hence **x** was selected as the vector that holds the node degrees.

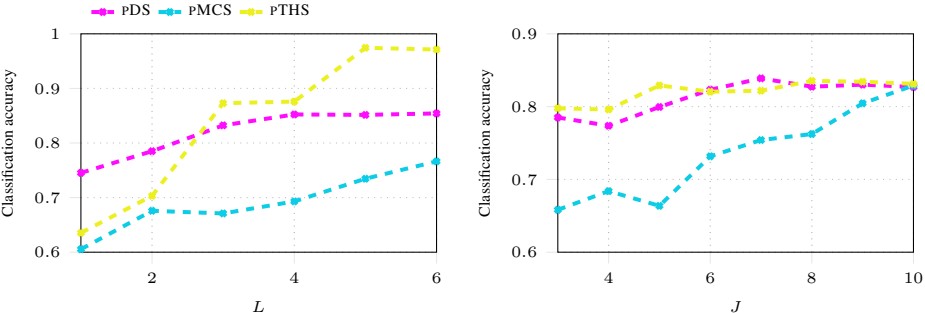

Figure 6: Classification accuracy over different parameters.

## A.1 ABLATION STUDY

Fig. 5 reports how the pGST is affected by varying the threshold $\tau$ in the task of source localization, with $J = 6$ and $L = 5$. Fig. 5 (a) shows the classification accuracy that generally decreases at $\tau$ increases since the number of active features $|\mathcal{T}|$ decreases; cf. Fig. 5 (b). Fig. 5 (c) reports the runtime in seconds of the approaches. Fig. 6 showcases the classification performance of the pGST with $\tau = 0.01$ for varying $L$ with $J = 3$ on the left and for varying $J$ with $L = 3$ on the right. It is observed, that the classification performance generally increases with $L$ and $J$.

## B PROOF OF THEOREM 1

First, the objective in (4) is rewritten as

$$\sum_{j=1}^{J} \left( \sum_{n=1}^{N} \left( \widehat{h}_j(\lambda_n)^2 - \tau \right) [\widehat{\mathbf{z}}_{(p)}]_n^2 \right) f_j = \sum_{j=1}^{J} \widehat{\mathbf{z}}_{(p)}^{\top} \left( \operatorname{diag} \left( \widehat{h}_j(\boldsymbol{\lambda}) \right)^2 - \tau \mathbf{I} \right) \widehat{\mathbf{z}}_{(p)} f_j \tag{10}$$

that follows by definition. By introducing the scalars $\alpha_j := \widehat{\mathbf{z}}_{(p)}^{\top} (\operatorname{diag}(\widehat{h}_j(\boldsymbol{\lambda}))^2 - \tau \mathbf{I}) \widehat{\mathbf{z}}_{(p)}$ for $j = 1, \ldots, J$, (4) can be rewritten as

$$\max_{f_j} \quad \sum_{j=1}^{J} \alpha_j f_j \tag{11}$$
$$\text{s. t.} \quad f_j \in \{0, 1\}, \quad j = 1, \ldots, J.$$

The optimization problem in (11) is nonconvex since $f_j$ is a discrete variable. However, maximizing the sum in (11) amounts to setting $f_j = 1$ for the positive $\alpha_j$ over $j$. Such an approach leads to the optimal pruning assignment variables as follows

$$f_j^* = \begin{cases} 1 & \text{if } \alpha_j > 0, \\ 0 & \text{if } \alpha_j < 0. \end{cases}, \quad j = 1, \ldots, J \tag{12}$$

The rest of the proof focuses on rewriting $\alpha_j$ as follows

$$\alpha_j = \widehat{\mathbf{z}}_{(p)}^{\top} (\operatorname{diag}(\widehat{h}_j(\boldsymbol{\lambda}))^2 - \tau \mathbf{I}) \widehat{\mathbf{z}}_{(p)} \tag{13}$$
$$= \|\operatorname{diag}(\widehat{h}_j(\boldsymbol{\lambda})) \widehat{\mathbf{z}}_{(p)}\|^2 - \tau \|\widehat{\mathbf{z}}_{(p)}\|^2 \tag{14}$$

Furthermore, since $\mathbf{V}$ is orthogonal matrix it holds that $\|\widehat{\mathbf{z}}_{(p)}\|^2 = \|\mathbf{V}^{\top} \mathbf{z}_{(p)}\|^2 = \|\mathbf{z}_{(p)}\|^2$ and it follows

$$\|\operatorname{diag}(\widehat{h}_j(\boldsymbol{\lambda})) \widehat{\mathbf{z}}_{(p)}\|^2 = \|h_j(\mathbf{S}) \mathbf{z}_{(p)}\|^2 \tag{15}$$
$$= \|\sigma(h_j(\mathbf{S}) \mathbf{z}_{(p)})\|^2$$
$$= \|\mathbf{z}_{(p,j)}\|^2 \tag{16}$$

where the second line follows since $\sigma(\cdot)$ is applied elementwise and does not change the norm.

## C PROOF OF LEMMA 1

By definition (3) it holds

$$\|\boldsymbol{\Phi}(\mathbf{x}) - \boldsymbol{\Phi}(\tilde{\mathbf{x}})\|^2 = \sum_{\ell=0}^{L} \sum_{p^{(\ell)} \in \mathcal{P}^{(\ell)}} |\phi_{(p^{(\ell)})} - \tilde{\phi}_{(p^{(\ell)})}|^2 \tag{17}$$

Hence, it is well motivated to bound each term of the sum in (17) as follows

$$|\phi_{(p^{(\ell)})} - \tilde{\phi}_{(p^{(\ell)})}| = |U(\mathbf{z}_{(p^{(\ell)})} - U(\tilde{\mathbf{z}}_{(p^{(\ell)})}))| \tag{18}$$
$$\leq \|U\| \|\mathbf{z}_{(p^{(\ell)})} - \tilde{\mathbf{z}}_{(p^{(\ell)})}\| \tag{19}$$

where (19) follows since the norm is a sub-multiplicative operator. Next we will show the following recursive bound

$$\|\mathbf{z}_{(p^{(\ell)})} - \tilde{\mathbf{z}}_{(p^{(\ell)})}\| = \|\sigma(h_{j^{(\ell)}}(\mathbf{S})\mathbf{z}_{(p^{(\ell-1)})}) - \sigma(h_{j^{(\ell)}}(\mathbf{S})\tilde{\mathbf{z}}_{(p^{(\ell-1)})})\| \tag{20}$$

$$\leq \|\sigma()\|\|h_{j^{(\ell)}}(\mathbf{S})\mathbf{z}_{(p^{(\ell-1)})} - h_{j^{(\ell)}}(\mathbf{S})\tilde{\mathbf{z}}_{(p^{(\ell-1)})}\| \tag{21}$$

$$\leq \|h_{j^{(\ell)}}(\mathbf{S})\mathbf{z}_{(p^{(\ell-1)})} - h_{j^{(\ell)}}(\mathbf{S})\tilde{\mathbf{z}}_{(p^{(\ell-1)})}\| \tag{22}$$

$$\leq \|h_{j^{(\ell)}}(\mathbf{S})\|\|\mathbf{z}_{(p^{(\ell-1)})} - \tilde{\mathbf{z}}_{(p^{(\ell-1)})}\| \tag{23}$$

where (21), (23) follow since the norm is a sub-multiplicative operator and (22) follows since the nonlinearity is nonexpansive, i.e. $\|\sigma()\| < 1$. Hence, by applying (23) $\ell - 1$ times the following condition holds

$$\|\mathbf{z}_{(p^{(\ell)})} - \tilde{\mathbf{z}}_{(p^{(\ell)})}\| \leq \|h_{j^{(\ell)}}(\mathbf{S})\|\|h_{j^{(\ell-1)}}(\mathbf{S})\|\cdots\|h_{j^{(1)}}(\mathbf{S})\|\|\mathbf{x} - \tilde{\mathbf{x}}\| \tag{24}$$

and by further applying the frame bound and (6) it follows that

$$\|\mathbf{z}_{(p^{(\ell)})} - \tilde{\mathbf{z}}_{(p^{(\ell)})}\| \leq B^{\ell}\|\boldsymbol{\delta}\| \tag{25}$$

Combining (19), (25) and the average operator property $\|U\| = 1$ it holds that

$$|\phi_{(p^{(\ell)})} - \tilde{\phi}_{(p^{(\ell)})}| \leq B^{\ell}\|\boldsymbol{\delta}\| \tag{26}$$

By applying the bound (26) for all entries in the right hand side of (17) it follows that

$$\|\boldsymbol{\Phi}(\mathbf{x}) - \boldsymbol{\Phi}(\tilde{\mathbf{x}})\|^2 \leq \sum_{\ell=0}^{L} \sum_{p^{(\ell)} \in \mathcal{P}^{(\ell)}} B^{2l}\|\boldsymbol{\delta}\|^2 \tag{27}$$

By factoring out $\|\boldsymbol{\delta}\|$ and observing that the sum in the right side of (27) does not depend on the path index $p$ it follows that

$$\|\boldsymbol{\Phi}(\mathbf{x}) - \boldsymbol{\Phi}(\tilde{\mathbf{x}})\|^2 \leq \left(\sum_{\ell=0}^{L} |\mathcal{P}^{(\ell)}| B^{2\ell}\right)\|\boldsymbol{\delta}\|^2 \tag{28}$$

Finally, since the cardinality of the paths at $\ell$ is $|\mathcal{P}^{(\ell)}| = J^{\ell}$ and $\sum_{\ell=0}^{L}(B^2 J)^{\ell} = \left((B^2 J)^L\right) / \left(B^2 J - 1\right)$ it holds

$$\|\boldsymbol{\Phi}(\mathbf{x}) - \boldsymbol{\Phi}(\tilde{\mathbf{x}})\| \leq \sqrt{\frac{(B^2 J)^L}{B^2 J - 1}}\|\boldsymbol{\delta}\| \tag{29}$$

## D    PROOF OF LEMMA 2

We will prove the case for $\ell = 0$, where $\mathbf{z}_{p^{(0)}} = \mathbf{x}$, since the same proof holds for any $\ell$. First, we adapt (8) to the following

$$\left|\|h_j(\mathbf{S})\mathbf{x}\|^2 - \tau\|\mathbf{x}\|^2\right| > \|h_j(\mathbf{S})\boldsymbol{\delta}\|^2 + \tau\left|\|\mathbf{x}\|^2 - \|\tilde{\mathbf{x}}\|^2\right|. \tag{30}$$

The proof will examine two cases and will follow by contradiction. For the first case, consider that branch $j$ is pruned in $\boldsymbol{\Psi}(\mathbf{x})$ and not pruned in $\boldsymbol{\Psi}(\tilde{\mathbf{x}})$, i.e. $(j) \in \mathcal{T}$ and $(j) \notin \tilde{\mathcal{T}}$. By applying (5) for $\mathbf{z}_{(j)} = \sigma(h_j(\mathbf{S})\mathbf{x})$ there exists $C \geq 0$ such that

$$\frac{\|h_j(\mathbf{S})\mathbf{x}\|^2}{\|\mathbf{x}\|^2} \leq \tau - C \tag{31}$$

$$\|h_j(\mathbf{S})\mathbf{x}\|^2 \leq \tau\|\mathbf{x}\|^2 - C\|\mathbf{x}\|^2 \tag{32}$$

Furthermore, from (5) it holds for $\tilde{\mathbf{z}}_{(j)} = \sigma(h_j(\mathbf{S})\tilde{\mathbf{x}})$ that

$$\frac{\|h_j(\mathbf{S})\tilde{\mathbf{x}}\|^2}{\|\tilde{\mathbf{x}}\|^2} > \tau \tag{33}$$

By applying (6) to (33), and using the triangular inequality it follows that

$$\|h_j(\mathbf{S})\mathbf{x}\|^2 + \|h_j(\mathbf{S})\boldsymbol{\delta}\|^2 \geq \tau\|\tilde{\mathbf{x}}\|^2 \tag{34}$$

Next, by applying (32) it holds that

$$\tau\|\mathbf{x}\|^2 - C\|\mathbf{x}\|^2 + \|h_j(\mathbf{S})\boldsymbol{\delta}\|^2 \geq \tau\|\tilde{\mathbf{x}}\|^2 \tag{35}$$

$$\tau(\|\mathbf{x}\|^2 - \|\tilde{\mathbf{x}}\|^2) + \|h_j(\mathbf{S})\boldsymbol{\delta}\|^2 \geq C\|\mathbf{x}\|^2. \tag{36}$$

Next, by utilizing (30) and the absolute value property $|a| \geq a$ to upper-bound the left side of (36) it follows that

$$\|h_j(\mathbf{S})\mathbf{x}\|^2 - \tau\|\mathbf{x}\|^2 > C\|\mathbf{x}\|^2. \tag{37}$$

Finally, by applying (32) the following is obtained

$$0 > 2C\|\mathbf{x}\|^2 \tag{38}$$

which implies that $C < 0$. However, this contradicts (31) since $C \geq 0$. Following a symmetric argument we can complete the proof for the other case.

