# OpenReview forum: "Pruned Graph Scattering Transforms"
_ICLR.cc/2020/Conference — Accept (Poster)_

### Official Review · AnonReviewer3 · 2019-10-23
**Official Blind Review #3**

**Rating:** 6

**Review:**

In this paper, the authors developed graph scattering transforms (GST) with a pruning algorithm, with the aim to reduce the running time and space cost, improve robustness to perturbations on input graph signal, and encourage flexibility for domain adaption. To this end, pruned graph scattering transform (pGST) was proposed based on the alignment between graph spectrum of the graph filters and the scattering feature. The intuition is to consider tree nodes as subbands in the graph spectrum and prune tree nodes that do not have sufficient overlap with the graph spectrum of a graph signal. The pruning problem was formulated as an optimization problem, and a solution was developed with theoretical analysis. Moreover, the analysis on the stability and sensitivity to perturbations were provided. Overall, the algorithm development is solid. The experimental results demonstrate the proposed pGST can outperform GST on graph classification task with less running time. Comparing with some supervised GNN methods, the proposed pGST can still achieve comparable results on several datasets.

First, it is good to see sufficient theoretical analysis on the optimization algorithm and algorithmic stability. However, one concern here is on the time complexity analysis. As mentioned in this work, the goal is to reduce running time and space cost, but the authors are aware that the worst case time complexity of pGST is the same as GST. Intuitively, with proper pruning, the efficiency can be improved, but it seems to be unknown on when will pGST achieve reasonable efficiency. More analysis, for example, on the impact of tau, may be helpful. In the experiments, it is better to see more insights on scattering pattern analysis, such as on the reasons behind the difference between using shallow GCN for social graphs and using deep GCN for SD point clouds. The current analysis only confirms previous results which didn't use pGST. Thus it is interesting to see what's more can be provided by using pGST to confirm these results. As a minor suggestion, the statistics of the graph data of authorship and facebook can be provided of better understanding of the results.


**Experience Assessment:**

I have read many papers in this area.

**Review Assessment: Checking Correctness Of Derivations And Theory:**

I assessed the sensibility of the derivations and theory.

**Review Assessment: Checking Correctness Of Experiments:**

I carefully checked the experiments.

**Review Assessment: Thoroughness In Paper Reading:**

I read the paper at least twice and used my best judgement in assessing the paper.

---

> ### Author Response · Authors · 2019-11-15
> **Response to AnonReviewer3**
>
> Q1. Intuitively, with proper pruning, the efficiency can be improved, but it seems to be unknown on when will pGST achieve reasonable efficiency. More analysis, for example, on the impact of tau, may be helpful.
> Graph scattering networks are an established way of encoding the structure of graphs. One of their drawbacks is that, due to having to consider all scattering paths (all possible combination of wavelets coming from each layer) their space complexity scales exponentially with the depth of the network.
>
> R1. Additional analysis was included in Appendix A1 to assess the impact of $\tau$ to the performance of pGST. Fig. 5 reports how the pGST is affected by varying the threshold $\tau$ in the task of source localization, with $J=6$ and $L=5$. Specifically, the classification accuracy, the number of retained scattering features and the runtime of pGST are reported for varying $\tau$.
>
> To further analyze the role of the hyperparameter $\tau$, the updated manuscript includes Appendix B, where a parallelism is indicated between the proposed pruning criterion and an optimal rate-distortion ratio. Upon appropriately selecting $\tau$, the pruning objective reduces to selecting the scattering features that minimize the global distortion for a given budget on the number of retained features (Ramchandran  and  Vetterli, 1993).
>
> Q2. In the experiments, it is better to see more insights on scattering pattern analysis, such as on the reasons behind the difference between using shallow GCN for social graphs and using deep GCN for SD point clouds. The current analysis only confirms previous results that didn't use pGST. Thus it is interesting to see what's more can be provided by using pGST to confirm these results.
>
> R2. In Appendix A.4, we performed additional experiments to empirically validate the insights from the scattering patterns.
>
> Fig. 8 depicts the pruned scattering patterns for the protein datasets namely Enzyme, Protein and DD. Evidently, most connections after $l=5$ are pruned, which suggests that scattering features obtained from up to 5 graph convolution layers are the most informative.
>
> Fig. 9 evaluates the performance of DiffPool (Ying et al., 2018) for these datasets as the number of GCN layers in the model increases. It is observed, that the performance of DiffPool does not improve significantly after 5 GCN layers. This finding corroborates the insights obtained from the pGSTs in Fig.8.
>
> These results motivate our future research that will focus on designing pGST-based GCNs were the network parameters will be judiciously selected based on the proposed pruning framework.
>
> Q3. As a minor suggestion, the statistics of the graph data of authorship and facebook can be provided of better understanding of the results.
>
> R3. The number of nodes is $N=234$ for Facebook graphs and $N=244$ for authorship graphs. These statistics are included in the updated experimental section.

---

### Official Review · AnonReviewer1 · 2019-10-23
**Official Blind Review #1**

**Rating:** 6

**Review:**

A scattering transform on graphs consists in the cascade of wavelets, modulus non-linearity and a low-pass filter. The wavelets and the low-pass are designed in the spectral domain, which is computationally extensive. Instead to compute any cascades of wavelets, this paper proposes to prune scattering paths which have the lowest energy. This is simple, and numerically efficient.

I find this idea elegant because it is simple, and it seems to work well(and even improve standard GST). Yet, I would have liked to see more explanations concerning the threshold selection: indeed, a naive simple grid search to find the best threshold could be less efficient than doing no pruning.(at least at training, but also for testing: maybe some guarantees that the test accuracy won't be lower under reasonable assumptions)

Pros :
- Good performances with a simple method. This is a positive aspect of the paper.
- This is an attempt to scale spectral methods on large graphs.
- A stability study of the practical impact of this threshold is done.

Cons:
- The bound in lemma 1 sounds pretty un-optimal. Indeed, the frame bound is B, thus the corresponding operator is B-lipschitz, and the modulus operator is 1-lipscthiz. Thus, shouldn't the lipschitz constant be simply $\sim B^{L+1}$? I don't understand why it does depend on the number of filters. Also, if B>1, then the lipschitz constant can be arbitrary high. This sounds to me like a big issue, because high lipschitz constant would indicate an unstability. For instance if $B\geq 1$, for order 1:
$B^4\Vert x\Vert^2\geq B^2\Vert Hx\Vert^2 + B^2\Vert Ux\Vert^2\geq \Vert |H||H|x\Vert^2+\Vert U|H|x\Vert^2+B^2\Vert Ux\Vert^2\geq \Vert \Phi x\Vert^2$
- How is the threshold selected? It is obvious that the normalisation of the filters play an extremely important role(if the frame was tight, I wouldn't complain). This should be well documented and this is currently not the case. I find this is critical. Maybe a parallel with the bias of soft-thresholding operators could be drawn to find an optimal value?
- I also believe the threshold shouldn't take in account the low-pass filter contribution. Indeed, as for equation (5), the energy of z_(p) is the sum of the energy of the high frequencies of z_p and the low frequency energy. It would not add any compute time(because this quantity is computed anyway) to measure the relative energy w.r.t. the high frequencies without the low-pass filter. It would write then: ||z_(puj)||^2/(||z_p||^2-||U(z_p)||^2) . Indeed, in classical signal processing (e.g., images, spectrogram), the low pass filter contains much more energy than other frequencies, which are however informative. In the current setting, discriminative scattering path could be discarded simply because the wavelet combined with a modulus did a good job at demodulating the signal to the low frequency domain.
- ". Furthermore, stable features are not necessary useful for learning. For example, an all-zero scattering feature at a certain output channel of the GST is non-informative albeit stable to any deformation" > I find this statement pretty obvious, could the authors think of a better example? In my opinion, stability to deformation at not extra-cost is always desirable because it helps creating a linear invariants.


minor:
- "Under certain conditions on graph filter banks, GSTs are endowed with energy conservation properties, as well as stability meaning robustness to graph topology deformations (Gama et al., 2019a)." > I think the citation about (Zou&Lerman) should be added here. (next to "conservation property")
- The notation p^{(l)}\cup j can be a bit confusing, couldn't this author write (p^{(l)},j) or anything similar?
- Top equation of page 5: $V^Th_j(S)x=diag(h_j)$. Shouldn't there be a \hat on diag(h_j)? Indeed, the current notation is a bit confusing... I guess it would be nice to introduce the convolution operator, in Fourier.
- "if the graph spectrum {hj (λn)}n is not aligned with [xb]n" . I'm not sure to understand the term "align" in this context, in particular because it seems to be used in different contexts in the paper. Could you simplify to "if the support of h_j is included in the support of \hat x?"
- Figures and Tables are too small...

post rebuttal:
I found the arguments of the authors convincing, they addressed my concern and revised the paper accordingly.

**Experience Assessment:**

I have published in this field for several years.

**Review Assessment: Checking Correctness Of Derivations And Theory:**

I carefully checked the derivations and theory.

**Review Assessment: Checking Correctness Of Experiments:**

I carefully checked the experiments.

**Review Assessment: Thoroughness In Paper Reading:**

I read the paper thoroughly.

---

> ### Author Response · Authors · 2019-11-15
> **Response to AnonReviewer1 1/2**
>
> Q1. The bound in lemma 1 sounds pretty un-optimal. Indeed, the frame bound is $B$, thus the corresponding operator is $B$-lipschitz, and the modulus operator is $1$-lipscthiz. Thus, shouldn't the lipschitz constant be simply $~B^{L+1}$ ? I don't understand why it does depend on the number of filters. Also, if $B>1$, then the lipschitz constant can be arbitrarily high. This sounds to me like a big issue, because high lipschitz constant would indicate an unstability. For instance if $B\ge1$, for order 1:
> $$
>     B^4\|\mathbf{x}\|^2\ge B^2\|\mathbf{H}\mathbf{x}\|^2+B^2\|\mathbf{U}\mathbf{x}\|^2\ge\||\mathbf{H}||\mathbf{H}|\mathbf{x}\|^2+
>     \||\mathbf{U}||\mathbf{H}|\mathbf{x}\|^2 +B^2\|\mathbf{U}\mathbf{x}\|^2\ge\|{\Phi}\mathbf{x}\|^2
> $$
> R1. The bound in (7) of Lemma 1 is not sub-optimal.' The misunderstanding may be due to the missing normalization by the number of scattering features $|{\Phi}(\mathbf{x})| = \sum_{\ell=1}^L{J}^\ell$ that we now include in the updated manuscript. The number of filters appears in (7) upon expanding the following norm as
> \[\|{\Phi}(\mathbf{x})-\Phi(\tilde{\mathbf{x}})\|^2
>     =\sum_{\ell=0}^{L} \sum_{p^{(\ell)}\in\mathcal{P}^{(\ell)}}
>     | \phi_{(p^{(\ell)})}-
>     \tilde{\phi}_{(p^{(\ell)})}|^2.\]
> Lemma 1 proceeds by bounding each summand of the norm, and hence the final bound was proportional to the number of terms  $\sum_{\ell=1}^L{J}^\ell$.
>
> Factors ${J}^\ell$ and ${B}^\ell$ also appear in stability bounds of related works; see e.g. eq. (19) in Theorem 1 of (Gama et al., NeurIPS 2019), and eq. (66) of (Zou and Lerman 2019) that pursue stability of the GST to perturbations in the graph structure.
>
> To enhance clarity, we normalize the bound in the updated manuscript by the square root of the number of scattering features. This alleviates the scaling of the bound with $J$. The updated bound reads as
> $$
>     \frac{\|{\Phi}(\mathbf{x})-\Phi (\tilde{\mathbf{x}})\|_2}{\sqrt{|{\Phi}(\mathbf{x})|}}  \le
>     \sqrt{\frac{\sum_{\ell=0}^{L}(B^{2}{J})^\ell}{\sum_{\ell=0}^L{J}^\ell}}
>     \|{{\delta}}\|_2~
> $$
> The dependency of the updated bound with respect to $B$ also reveals the importance of choosing tight frames having $B=1$. If the frame is tight, the resulting normalized stability bound is also tight. Finally, although the derivations and proofs consider general wavelets, tight wavelets are preferable, and are used in the experiments. Lemma 1, Theorem 2 and the paragraphs following them, were updated to clarify this point.
>
> Q2. How is the threshold selected? It is obvious that the normalization of the filters plays an extremely important role(if the frame was tight, I wouldn't complain). This should be well documented and this is currently not the case. I find this is critical. Maybe a parallel with the bias of soft-thresholding operators could be drawn to find an optimal value?
>
> R2. Our experiments include tight Hann wavelets; see also (Shuman et al., 2015). Cross-validation is performed to select $\tau$ over a grid with 10 points. Note also that the competing approaches, especially those based on graph neural networks, have to perform cross-validation over a significantly larger parameter space. To establish the sensitivity of pGST with respect to $\tau$, an ablation study was performed in Fig. 5 of Appendix A1. Fig. 5 reports the classification accuracy, the number of retained scattering features, and the runtime of pGST across $\tau$ values.
>
> As suggested, a parallelism is indicated between the proposed pruning criterion, and an optimal rate-distortion ratio. Upon appropriately selecting $\tau$, the criterion in (5) yields the optimal variables in a rate-distortion sense; see K. Ramchandran and M. Vetterli, ``Best wavelet packet bases in a rate-distortion sense,'' 1993. These variables are given by solving the following problem
> $$
> \max_{\{ f_{(p, j)} \}_{j=1}^J}~
>     \sum^{J}_{j=1}\frac{
> \|{\mathbf{z}}_{(p,j)}\|_2^2}
> {\|{\mathbf{z}}_{(p)}\|_2^2}f_{(p, j)}~\text{s. t.}~~~~~ f_{(p, j)} \in \{0,1\}~\forall j, \text{     }  \sum^{J}_{j=1}f_{(p, j)} \le K$$
> where $K$ is the maximum number of retained scattering features, as dictated by computational complexity constraints. Indeed, by properly choosing $\tau$, the criterion in (5) can be employed to find the $K$ channels with maximum energy. A similar energy preservation objective has been employed in the signal processing literature in the context of PCA wavelets to achieve minimum distortion; see M. K. Tsatsanis and G. B. Giannakis, ``Principal component filter banks for optimal multiresolution analysis,'' 1995. Hence, the pruning objective boils down to selecting the scattering features that minimize the global distortion for a given budget $K$ (Ramchandran and Vetterli, 1993).
>
> The connection is further elaborated in the updated Appendix B, to enhance clarity on the role of the hyperparameter $\tau$.

---

> > ### Author Response · Authors · 2019-11-15
> > **Response to AnonReviewer1 2/2**
> >
> > Q3. I also believe the threshold shouldn't take into account the low-pass filter contribution. Indeed, as for eq. (5), the energy of $z_{(p)}$ is the sum of the energy of the high frequencies of $z_{(p)}$ and the low frequency energy. It would not add any compute time(because this quantity is computed anyway) to measure the relative energy w.r.t. the high frequencies without the low-pass filter. It would write then: $||z_{(puj)}||^2/(||z_p||^2-||U(z_p)||^2)$. Indeed, in classical signal processing (e.g., images, spectrogram), the low pass filter contains much more energy than other frequencies, which are however informative. In the current setting, discriminative scattering path could be discarded simply because the wavelet combined with a modulus did a good job at demodulating a signal to the low frequency domain.
> >
> > R3. The pruning criterion suggested by the reviewer compares the energy at the filter output with the energy only of the highpass components at the input. Such a design retains all scattering features from lowpass filters and more features from highpass filters, since the energy in the denominator is smaller. On the other hand, the pruning criterion in the submitted manuscript directly maximizes the retained energy after pruning.
> >
> > Additional experiments are included in Appendix A.2 that compare the pruning criterion advocated in this paper (pGST) and the one proposed by the reviewer (abbreviated as pGSTalt) in the task of source localization in the Facebook graph. Fig. 7 shows that pGST and pGSTalt perform very similarly in terms of classification accuracy, while pGSTalt retains significantly more features and hence the runtime of pGSTalt is several orders of magnitude higher runtime than that of pGST.
> >
> > Finally, Fig. 3 (d) depicts the percentage of features after pruning retained in the top most discriminative GST features as given by the SVM classifier.  It is observed, that although pGST does not take into account the labels, the retained features are indeed discriminative for classification.
> >
> > The updated Appendix A.2 analyzes this high-pass promoting criterion and compares it experimentally with the one proposed in the submitted manuscript.
> >
> > Q4. ``Furthermore, stable features are not necessarily useful for learning. For example, an all-zero scattering feature at a certain output channel of the GST is non-informative albeit stable to any deformation.'' I find this statement pretty obvious, could the authors think of a better example? In my opinion, stability to deformation at not extra-cost is always desirable because it helps create linear invariants.
> >
> > R4. Thanks for pointing this out. Indeed, stable features are desirable and our work pursues establishing stability to perturbations. Stability, however, should not come at odds with sensitivity. A filter’s output should be sensitive to and ``detect’’ perturbations of large magnitude. The sensitivity of pGST is established in Appendix C. As suggested, we updated the indicated statement to convey this message.
> >
> > Q5. Minor comments.
> >
> > R5. Thank you for the detailed comments. The suggested points are updated and the figures enlarged. If the paper is accepted, the figures and tables will be further enlarged using the additional space.

---

> > > ### Comment · AnonReviewer1 · 2019-11-15
> > > **2/2**
> > >
> > > Q3: Thank you. I understand better why the method works.
> > >
> > > Q4: Thanks.
> > >
> > > Q5: Thanks.

---

> > ### Comment · AnonReviewer1 · 2019-11-15
> > **Thank for your rebuttal**
> >
> > Dear author, thanks for your reply.
> >
> > Q1: I think dividing the representation by its lipschitz constant is not a fair argument, as then, $\Vert x\Vert =1$, then $\Vert \Phi x\Vert\sim J^{-L-1}$ and the representation is not close to be unitary(in particular if $B=1$)... this induces huge renormalization effect.
> >
> > Q2: Thanks for the clarification.

---

> > > ### Author Response · Authors · 2019-11-15
> > > **Thanks for the update**
> > >
> > > Thanks for the immediate reply and the useful feedback.
> > >
> > > The normalization is applied only to the bound to alleviate the expected scaling of the error with the number of elements in $\Phi{\mathbf{x}}$. Graph scattering transforms introduce a redundant representation for a graph signal $\mathbf{x}$. The redundancy increases as the number of scattering layers grows. It is then not a fair comparison among graph scattering transforms with a different number of layers and hence different levels of redundancy. After normalization, graph scattering transforms still have redundant basis vectors, but each one has a smaller norm.
> > >
> > > Without any normalization, this redundant representation is indeed not close to be unitary. However, this is due to the property of the graph scattering transform, not due to a loose bound.
> > >
> > > The factors $B^l$, $J^l$ also appear in related works (Gama et al., NeurIPS 2019) and (Zou and Lerman 2019) and are a consequence of the exponential number of scattering features in GSTs.
> > >
> > > Although the proofs of this paper consider general wavelets for completeness, tight wavelets with $B=1$ are used in practice.

---

### Official Review · AnonReviewer2 · 2019-11-08
**Official Blind Review #2**

**Rating:** 6

**Review:**

Graph scattering networks are an established way of encoding the structure of graphs. One of their drawbacks is that, due to having to consider all scattering paths (all possible combination of wavelets coming from each layer) their space complexity scales exponentially with the depth of the network.

The paper proposes a simple scheme for pruning the contributing wavelets so as to retain only those that are best aligned with the input signal of the given layer. The experiments show that this leads to scattering networks that perform about equally well, but are much sparser.

It is not entirely clear why alignment with the input signal should be a good proxy for how informative a given wavelet is. Depending on what downstream algorithm the scattering network's output is fed into, sometimes small coefficients could be important too. It might be unfair to only use an SVM in the experiments, since for SVMs larger magnitude features are important more likely to be relevant.

Some stability results are presented, but they are not terribly deep, and do not address the fact that the optimization is problem is highly nonlinear and therefore even after relatively small perturbations drastically different sets of wavelets may be selected. Nonetheless, this is an interesting contribution to the field of graph scattering networks.

**Experience Assessment:**

I have read many papers in this area.

**Review Assessment: Checking Correctness Of Derivations And Theory:**

I assessed the sensibility of the derivations and theory.

**Review Assessment: Checking Correctness Of Experiments:**

I assessed the sensibility of the experiments.

**Review Assessment: Thoroughness In Paper Reading:**

I read the paper thoroughly.

---

> ### Author Response · Authors · 2019-11-15
> **Response to AnonReviewer2**
>
> Q1. It is not entirely clear why alignment with the input signal should be a good proxy for how informative a given wavelet is. Depending on what downstream algorithm the scattering network's output is fed into, sometimes small coefficients could be important too. It might be unfair to only use an SVM in the experiments since for SVMs larger magnitude features are important more likely to be relevant.
>
> R1. SVM is not the only downstream classifier used in the experiments. For the graph classification tests in Table 1, the gradient boosting classifier is employed, while for point cloud classification, a fully connected neural network is used. The SVM classifier is only used in Fig. 3, and the reason was to reproduce the experiments of  (Gama et al., NeurIPS 2019) and compare our method in the same footing.
>
> The experiments show that the spectrum alignment is indeed a good proxy for pruning. The proposed pGST that only uses the pruned features, matches and often outperforms the full GST in Figs. 3,4 and Table 1. Furthermore, Fig. 3 (d) depicts the percentage of features after pruning retained in the top most discriminative GST features as given by the SVM classifier.  It is observed, that although pGST does not take into account the labels, the retained features are indeed discriminative for classification.
>
> To further clarify the intuition behind our pruning criterion, Appendix B includes additional analysis connecting the proposed criterion to a minimum distortion objective. Upon appropriately selecting $\tau$, the pruning objective is equivalent to selecting the scattering features that minimize the global distortion for a given budget on the number of retained features (Ramchandran  and  Vetterli, 1993).
>
> Q2. Some stability results are presented, but they are not terribly deep, and do not address the fact that the optimization is problem is highly nonlinear and therefore even after relatively small perturbations drastically different sets of wavelets may be selected. Nonetheless, this is an interesting contribution to the field of graph scattering networks.
>
> R2. The nonconvexity of the optimization problem in (4) is directly addressed by Theorem 1, which states that the optimum of (4) is achieved by selecting the optimization variables as in (5). When the input is perturbed, Lemma 2 establishes that the sets of selected wavelets are the same given that the SNR condition in (9) holds.

---

### Author Response · Authors · 2019-11-15
**Summary of Changes**

We uploaded the revised manuscript.

Summary of updates.

-Normalized the bound in Lemma 1 by the number of elements in ${\Phi}(\mathbf{x})$.
-Included a parallelism between the proposed pruning objective, and a minimum distortion criterion.
-Performed additional experiments with an alternative pruning criterion.
-Analyzed additional pruned scattering patterns for protein datasets, and explored how these patterns relate to the effective depth of GCNs.

We thank all reviewers for providing valuable feedback that has certainly improved the quality of the manuscript.

---

### Decision · Program_Chairs · 2019-12-19

**Decision:**

Accept (Poster)

**Comment:**

Main content: Authors developed graph scattering transforms (GST) with a pruning algorithm, with the aim to reduce the running time and space cost, improve robustness to perturbations on input graph signal, and encourage flexibility for domain adaption.
Discussion:
reviewer 1: likes the idea, considers it to be elegant and work well. some questions regarding the proofs in the paper but it sounds like authors have addressed concerns.
reviewer 2: solid paper and results, has questons on stability results, like reviewer 2.
reviewer 3: likes the idea, including good sufficient theoretical analysis and algorthmic stability. concern is around complexity analysis but sounds like the authors have addressed the concerns.
Recommendation: Well written solid paper with good proofs. Authors addressed any reviewer concerns and all 3 reviewres vote weak accept. This is good for poster.